# Ionization of Xenon Clusters by a Hard X-ray Laser Pulse

Yoshiaki Kumagai [1,2,*], Weiqing Xu [1,3], Kazuki Asa [4], Toshiyuki Hiraki Nishiyama [4], Koji Motomura [1], Shin-ichi Wada [5,6], Denys Iablonskyi [1], Subhendu Mondal [1], Tetsuya Tachibana [1], Yuta Ito [1], Tsukasa Sakai [4], Kenji Matsunami [4], Takayuki Umemoto [5], Christophe Nicolas [7], Catalin Miron [8], Tadashi Togashi [9], Kanade Ogawa [6], Shigeki Owada [6], Kensuke Tono [9], Makina Yabashi [6], Hironobu Fukuzawa [1,6], Kiyonobu Nagaya [4,6] and Kiyoshi Ueda [1,6]

[1] Institute of Multidisciplinary Research for Advanced Materials, Tohoku University, Sendai 980-8577, Japan
[2] Department of Applied Physics, Tokyo University of Agriculture and Technology, Koganei-shi, Tokyo 184-8588, Japan
[3] Center for Transformative Science, ShanghaiTech University, Shanghai 201210, China
[4] Department of Physics, Kyoto University, Kyoto 606-8502, Japan
[5] Graduate School of Advanced Science and Engineering, Hiroshima University, Higashi-Hiroshima 739-8526, Japan
[6] RIKEN SPring-8 Center, Sayo 679-5148, Japan
[7] Synchrotron SOLEIL L'Orme des Merisiers Départementale 128, 91190 Saint-Aubin, France
[8] Université Paris-Saclay, CEA, CNRS, LIDYL, 91191 Gif-sur-Yvette, France
[9] Japan Synchrotron Radiation Research Institute (JASRI), Sayo 679-5198, Japan
[*] Correspondence: kumagai@go.tuat.ac.jp

**Abstract:** Ultrashort pulse X-ray free electron lasers (XFFLs) provided us with an unprecedented regime of X-ray intensities, revolutionizing ultrafast structure determination and paving the way to the novel field of non-linear X-ray optics. While pioneering studies revealed the formation of a nanoplasma following the interaction of an XFEL pulse with nanometer-scale matter, nanoplasma formation and disintegration processes are not completely understood, and the behavior of trapped electrons in the electrostatic potential of highly charged species is yet to be decrypted. Here we report the behavior of the nanoplasma created by a hard X-ray pulse interacting with xenon clusters by using electron and ion spectroscopy. To obtain a deep insight into the formation and disintegration of XFEL-ignited nanoplasma, we studied the XFEL-intensity and cluster-size dependencies of the ionization dynamics. We also present the time-resolved data obtained by a near-infrared (NIR) probe pulse in order to experimentally track the time evolution of plasma electrons distributed in the XFEL-ignited nanoplasma. We observed an unexpected time delay dependence of the ion yield enhancement due to the NIR pulse heating, which demonstrates that the plasma electrons within the XFEL-ignited nanoplasma are inhomogeneously distributed in space.

**Keywords:** hard X-ray laser; xenon cluster; nanoplasma; electron spectroscopy; ion time-of-flight spectroscopy; pump–probe experiment

## 1. Introduction

The recent availability of brief and intense X-ray free electron laser (XFEL) [1–3] pulses had a significant impact on such research areas as structural biology [4–6], transient species dynamics [7–9], and dynamic imaging of matter [10–12]. However, a prerequisite to undoubtedly establish and routinely implement these new experimental methodologies is understanding the reactions, and the possible radiation damage induced at the atomic level by the interaction of the XFEL pulses with the target. In single-shot diffraction experiments of amorphous particles, for example, photo-emissions and subsequent Auger emissions are known to cause damage of samples [13].

To gain insight into the XFEL-induced sample damage at the atomic level, numerous studies have been carried out on atoms [14–16] and molecules [17–22]. In the case of

atomic samples, multiple cycles of photo-emission and Auger decay lead to the ejection of multiple electrons resulting in a dramatic change in the electronic structure [14–16]. In the case of molecular samples, the multiply charged molecular ions resulting from the photoemission and Auger decay cycles explode violently due to the Coulomb repulsive force [17–23]. These fundamental processes at the atomic level are indeed likely to be responsible for the observed radiation damage, but the size of the molecules studied, consisting of about a dozen atoms or less, are still relatively small compared to macromolecules and nanostructured samples.

To investigate XFEL-ignited dynamics beyond the above simple targets, atomic clusters are ideal systems since their sizes are easily altered from a single atom to a bulk-like macroscopic region by employing a well-established method. Consequently, atomic clusters have been investigated extensively [24–27]. When an intense X-ray laser-pulse shines into an atomic cluster, numerous individual atoms are core-ionized, and many secondary electrons are emitted by the following Auger decays. The net charge of the ionized cluster increases due to the escape of electrons from the system, the Coulomb potentials of which electrically capture the other electrons after their kinetic energies are lost by inelastic scattering with atoms and ions in the system. A nanoplasma is thus formed [24,26]. This is expected to be a general phenomenon in XFEL-ignited dynamics, as the nanoplasma formation may be induced on any kind of nanometer-sized particles containing heavy elements. As such, understanding the formation and disintegration processes of the XFEL-ignited nanoplasma is not only of fundamental interest, but also of critical importance for the use of XFEL pulses for structural determinations [28,29].

Our recent study [30] demonstrated that the dedicated molecular dynamics simulation tool, XMDYN, captured essential physical processes steering the XFEL-ionization dynamics of argon clusters. When the clusters are irradiated with hard X-ray lasers of a moderate fluence, the ionization dynamics have been markedly affected by chemical processes, distributing the increasing net charge within the clusters and causing bond-reorganization, which induces the formation of oligomer fragments [30]. These dynamics induced by the moderate-fluence X-ray lasers are different from those by high-fluence ones where the system entirely decomposes into atomic species. Moderate-fluence XFEL pulses are expected to provide the ability to modify and transform the molecular structure of nanometer-sized objects by achieving control over the disintegration dynamics of XFEL-ionized clusters.

The XMDYN reported that on krypton clusters, the deep Coulomb potential effectively trapped the ionized electrons when its cluster size was greater than 5000 atoms [31]. Thermal electron emissions from the transient nanoplasma are highly suppressed owing to its deep Coulomb potential during the long-term plasma expansion associated with electron–ion recombination events. The simulation results show that this characteristic is likely to be observed in a wide range of sizes for relatively large-sized clusters. In other words, to decrypt the nanoplasma formation and disintegration processes, it is worth investigating the XFEL-ignited dynamics on small-sized clusters.

For a deeper insight into the dynamics of laser-ignited nanoplasmas, a number of time-resolved experimental studies have been conducted [32–37] using near-infrared (NIR) probe pulses. A recent study [34] demonstrated one novel concept that enables us to completely decouple the generation of seed electrons from the ionization dynamics driven by the NIR laser, both temporally and spatially. The study reported that inverse bremsstrahlung (IBS) heating, subsequent efficient avalanching by NIR pulses, and resonant excitation remain possibly well below the threshold of tunnelling ionization and can be activated by just several electrons seeded by an ultrashort extreme-ultraviolet (XUV) pulses produced by high-harmonic generation. Similar NIR-driven ionization dynamics has been observed on the XFEL-ignited nanoplasma containing ∼6 trapped electrons on average [37].

Ref. [35] reported another NIR-driven process, namely, surface plasma resonance [38,39] in high-density nanoplasma created by XUV-laser pulses. Surface plasma resonance causes an effective energy absorption of nanoplasma, which can be mathematically tantamount to a Mie resonance when the collective motion frequency of plasma electrons coincides

with the frequency of the NIR laser [40]. The surface plasma resonance effect has been also observed in the XFEL-ignited nanoplasma [31,36]. Further probing with NIR pulses has been expected to track the time evolution of the electron density within the XFEL-ignited nanoplasma.

Here, we report on electron and ion spectroscopies for xenon clusters exposed to X-ray laser pulses with the photon energy of 5.5 keV at the XFEL facility SACLA [3]. First of all, we focused on the dependence of ion yields produced from the XFEL-ignited nanoplasma on the XFEL intensity to obtain a deeper insight into the cluster dynamics with hard X-ray lasers of a moderate fluence. The XFEL fluence dependence indicates that the increase in the ionic mean charge state within the XFEL-ignited nanoplasma has hindered oligomer formation due to reorganization of the bonding. Moreover, to study the ionization dynamics of small-sized clusters, which has never been experimentally reported in the hard X-ray regime, we investigated the cluster-size dependence of ion data. The ion data for different-size clusters indicate that the oligomer formation is enhanced with the increase in cluster size. Furthermore, in order to deepen this insight, we performed time-resolved spectroscopy by using a probe NIR laser to track the expanding XFEL-ignited nanoplasma. The time-resolved data provide evidence that the trapped electrons may be inhomogeneously distributed in the XFEL-ignited nanoplasma produced from xenon clusters with an average size of $\sim$5000 atoms when the XFELs fulfill the moderate fluence condition.

## 2. Experiment

The experiments were performed at experimental hutch EH3 of beamline BL3 [41] of SACLA [3]. The apparatus used has been reported in detail elsewhere [37]. The photon energy and the photon bandwidth of X-ray pulses were 5.5 keV and $\sim$33 eV (full width at half maximum; FWHM), respectively. The XFEL pulse frequency was 30 Hz. The X-ray pulse has been characterized by its temporal duration that is shorter than ten femtoseconds [42]. A Kirkpatrick–Baez (KB) mirror arrangement focused the X-ray laser on the interaction point [43]. The focused beam size of the X-ray beam was measured to be $\sim$1 μm (FWHM). A beam-position monitor [44], which was located upstream of the KB mirror arrangement and calibrated by a calorimeter [45], recorded pulse energies of XFELs shot-to-shot, and the recorded fluctuation was $\pm$11% (22% FWHM). At the interaction point, the XFEL peak fluence was estimated to be $\sim$4.6 μJ/μm$^2$ on average using a calibration procedure employing yields of oligomer fragments from ionized argon clusters [30]. The intensity of the X-ray pulses was attenuated by aluminum foils located upstream on the KB mirror arrangement. The transmittances of the aluminum foils with thicknesses of 25 and 50 μm were 38% and 15%, respectively.

The clusters were produced by an adiabatic expansion of xenon gas through a 250 μm nozzle at room temperature. The stagnation pressures were 0.24, 0.32, and 0.65 MPa. The averaged cluster sizes, <*N*>, were estimated to be 500, 1000, and 5000 atoms, respectively, according to the scaling law [46]. The radii of clusters were 21, 26, and 44 Å, respectively. The uncertainty of the averaged cluster size was estimated to be $\pm$20% by the fluctuation of the stagnation pressure. Two skimmers, whose inner diameters were 0.5 and 2 mm, respectively, skimmed the cluster beam at 20 and 400 mm from the nozzle. The second skimmer was located 250 mm from the interaction point. The cluster beam perpendicularly intersects with the XFEL beam. At the interaction point, the transverse length of the cluster beam was estimated to be $\sim$2 mm (FWHM), i.e., shorter than the Rayleigh lengths of XFELs ($\sim$8 mm). Thus, electrons and ions were yielded from a cylindrical space with a diameter of $\sim$1 μm and a height of $\sim$2 mm along the propagation direction of the X-ray beam.

To probe the XFEL-ionized clusters, we employed NIR laser pulses with 800 nm wavelength. The focal size and pulse duration of the NIR laser were 200 μm (FWHM) and 82 fs (FWHM), respectively. The X-ray and NIR lasers overlapped at the interaction point with a small angle (<1°). The intensity of the NIR laser was adjusted using a neutral density

filter up to $5.0 \times 10^{12}$ W/cm$^2$, which was too weak to ionize the ground state of xenon atoms. Our experimental results supported this conclusion (not shown).

A temporal jitter between the XFEL and NIR pulses significantly restricts the intrinsic temporal resolution of the pump–probe method, determined by the temporal durations of the two pulses [47]. We did not record the arrival times of the XFEL and NIR laser pulses and thus assumed that the temporal resolution was equivalent to the typical temporal jitter between the pump–probe pulses (~800 fs (FWHM)) [36]. The scheme of pump–probe experiment is described in detail elsewhere [37].

Kinetic energy spectra of electrons and ionic fragments as a function of the pump–probe time-delay were measured by employing electron velocity map imaging (VMI) and ion time-of-flight (TOF) spectrometers [48], respectively. From the flight time and detected position ($X$, $Y$), the kinetic energies of ions were obtained. Here, we defined the $X$ and $Y$ axes as being parallel to the polarization direction and the propagation of the XFEL beam, respectively. In the calculation of kinetic energies, we did not distinguish the xenon isotopes; however, we used the average mass of 131.3 amu at the expense of energy resolution. In the present voltage setting, the electric fields worked in the VMI mode for ions emitted to the detector direction (up to 90 degrees from the spectrometer axis). However, the kinetic energy range of ions to be accepted up to 90 degrees is limited. We used a method we developed for collection angle compensation to obtain accurate ion kinetic energy distributions [48].

### 3. Results and Discussions

*3.1. Electron Emission in Atomic Processes*

First of all, an outline of the ionization processes for individual atoms is shown as a premise of the discussion about the ionization dynamics of clusters. In the 5.5 keV X-rays, we can classify photo- and Auger electrons derived from xenon atoms into two groups based on their initial kinetic energy. We have defined slow and fast groups as the electrons whose kinetic energies are initially below 1.2 keV and above 2.4 keV, respectively.

The deepest shell for photo-ionization of xenon atoms in the 5.5-keV X-rays is the $L$-shell, which is the most probable photo-ionization channel. Since the photo-absorption cross-section for xenon atoms at 5.5-keV of photon energy is 0.166 Mbarn [49] in the X-ray beam with the peak fluence of ~4.6 µJ/µm$^2$ the photo-absorption probability is approximately 8.7%, which denotes that the probability of multi-photon absorption is rather small, although for a single atom. A xenon cluster with the size of 5000 atoms absorbs ~430 photons, for example.

The percentages of the photo-ionization probabilities for $L_1$, $L_2$, and $L_3$ orbitals are 14%, 25%, and 45%, respectively [50]. The kinetic energies of electrons emitted from $L_1$, $L_2$, and $L_3$ photo-ionization by the 5.5-keV photons are ~47 eV, ~400 eV, and ~720 eV, respectively, which are slow electrons, as defined above. The other photo-ionization processes release fast electrons. Meanwhile, ~43% of $L_1$- and ~12% of $L_2$-vacancies produce slow electrons via Coster–Kronig transitions (LLN and LLO) [51,52]. $L$-holes are predominantly filled by electrons from shallow orbits in another Auger process releasing a fast electron (~51% of $L_1$, ~80% of $L_2$, ~93% of $L_3$ holes). In addition, the relaxation of $M$-, $N$-, and $O$-holes via Coster–Kronig transitions and Auger decays lead to the ejection of one slow electron. The mean charge state of xenon ions produced by 5.5 keV photons is ~7.7 [53]. Finally, 6.8 slow and 0.9 fast electrons are ejected from individual atoms on average by the photo-ionization and subsequent Auger cascades.

Collisional ionization is one of the dominant ionization events in a dense atomic environment, in which energetic free electrons interact with surrounding atoms to kick out a bound atomic electron. The secondary electrons produced by the collisional ionization event are typically categorized into the slow group.

The mean free path, $\lambda = (\sigma n)^{-1}$, is usually employed to explain the collisional ionization events in a medium, where $\sigma$ is the cross-section of collisional ionization and $n$ is the atomic density of the material. In a pristine xenon cluster, whose $n$ is $\sim 1.36 \times 10^{28}$ m$^{-3}$),

$\lambda$ becomes the shortest ($\sim$9.2 Å) at $\sim$70 eV kinetic energy and monotonically longer with an increase in the kinetic energy [54].

At the kinetic energy of 1.2 and 2.4 keV, which represent the upper limit of the slow electrons and the lower limit of the fast ones, respectively, their mean free paths are $\sim$39 and $\sim$82 Å, respectively, which are shorter and longer than the radius of a 5000-atom cluster (44 Å). In other words, the fast electrons do not create secondary electrons by the collisional ionization of xenon atoms in the cluster.

### 3.2. Electron Spectrum of Xenon Clusters Irradiated with XFEL Pulses

To discuss the involved photo-emission and the subsequent Auger emission, which are the initial stages of cluster ignition by irradiation of hard X-ray pulses, we focus on the electron energy spectrum. The black curve in Figure 1a depicts the electron energy spectrum of xenon clusters with an average size of 5000 atoms, irradiated with an XFEL pulse with the photon energy of 5.5 keV and the peak fluence of $\sim$1.8 µJ/µm$^2$. The electron spectrum exhibits four visible peaks and an unstructured contribution. The unstructured feature in the electron spectrum can be represented by the sum of two exponential functions, i.e., the high-energy contribution from $\sim$100 to $\sim$880 eV and the shoulder-like structure below $\sim$100 eV can be fitted by exponential decay functions with decay constants of 490 and 33 eV, respectively (see the red and green solid lines).

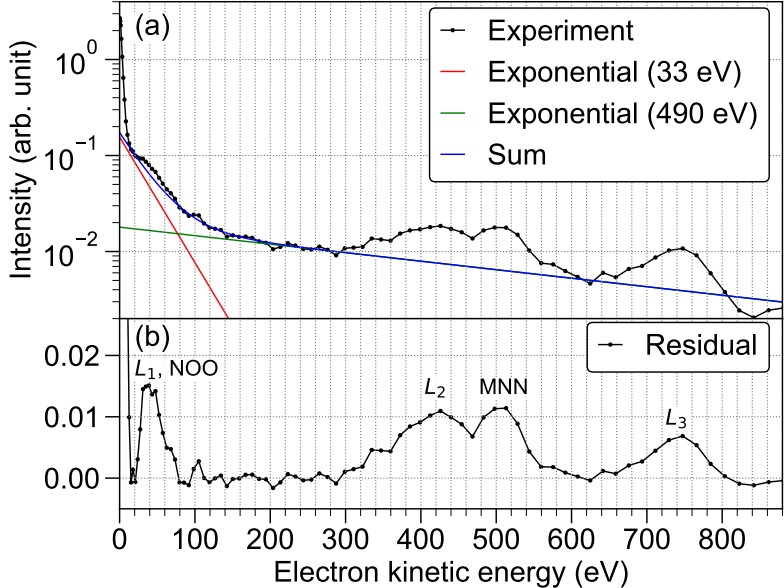

**Figure 1.** (**a**) Electron energy spectrum from Xe$_{5000}$ clusters at the XFEL peak fluence of 1.8 µJ/µm$^2$. Red and green solid lines depict exponential functions with decay constants of 33 and 490 eV, respectively. A blue solid curve shows the sum of the red and green lines. (**b**) Residual spectrum between the experimental results and the blue solid curve.

With the present peak fluence of the XFEL pulse ($\sim$1.8 µJ/µm$^2$), the ionization probability of an individual xenon atom is $\sim$3.3%, and thus, $\sim$170 atoms in the cluster of $\sim$5000 atoms are core-ionized. Therefore, $\sim$150 fast and $\sim$1100 slow electrons are emitted from these core-ionized atoms. Here, it is simply assumed that fast electrons entirely fly away from the cluster but slow ones do not. Then, the charge of the spherical cluster with a radius of 44 Å is initially assumed to be +150, and the electrostatic potential on the surface is 49 eV. This provisional electrostatic potential vindicates the simple assumption of ejecting fast electrons from the system but trapping slow ones, which is further supported by considering that the collisional ionization events additionally consume the kinetic energies of slow electrons.

When the charge of clusters is +150, the available electron temperature is expected to be below ~49 eV according to the electrostatic potential on the surface of $Xe_{5000}$. Therefore, the exponential feature with a decay constant of 33 eV (green solid line in Figure 1a) may be attributed to the thermal electron emissions from the XFEL-ignited nanoplasma. On the other hand, the other extracted decay constant (490 eV) is much higher than the expected electrostatic potential, e.g., an electrostatic potential of 490 eV requires a +2300 charge on $Xe_{5000}$. A number of slow electrons (<1.2 keV) can escape from the charged cluster after losing their kinetic energies via a sequence of collisional ionization events and may contribute to the exponentially decaying feature with a constant of 490 eV. The recent XMDYN simulation for krypton clusters [31] presented proof of the contribution of slow electrons to the exponentially decaying feature on the electron spectrum.

A number of photo- and Auger electrons, which are decelerated via collisional ionization processes, have been observed as the peaks. Figure 1b depicts a residual spectrum between the experimental electron spectrum and the sum of two exponential decay curves (see a blue curve in Figure 1a). In the previous electron spectroscopy study of xenon clusters with an average size of 10,000 atoms, the contributions of decelerated photo- and Auger electrons were also reported as evidence of the formation of nanoplasma by XFEL irradiation [55]. $L_2$ and $L_3$ photo-emissions, whose kinetic energies are originally ~400 and ~720 eV, respectively, in atomic case, have been observed as peaks, with a tail toward the low-energy side; this indicates that their kinetic energies are lost via the collisional ionization processes. The peak around 500 eV corresponding to MNN Auger electrons [56] also shows a tail due to the deceleration caused by the collisional ionization processes.

The electrons with the kinetic energy below the electrostatic potential of charged clusters cannot escape after the cluster charge increases. Thus, one can consider that while the NOO Auger decays, which emit ~50 eV electrons as the latter processes of Auger cascades, have minor contributions to a sharp peak around 50 eV, the $L_1$ photo-emissions have the main contributions to it.

The fact that the photo- and Auger electrons, which have been grouped as the slow electrons, have been observed as the peaks on the electron energy spectrum provides the evidence that a number of slow electrons can escape from the electrostatic potential of charged clusters during collisional ionization events. As a consequence, a number of the slow electrons escape from the system and in turn contribute to increasing the cluster charge. In other words, the cluster charge of +150 is a lower limit that has been estimated by the number of fast electrons produced under the present experimental conditions.

A strong emission of low-energy electrons (~0 eV) can be attributed to the contribution due to the efficient re-ionization of high-lying Rydberg states by a static electric field supplied by the electron VMI spectrometer [31,57]. The formation of Rydberg atoms indicates that the electron and ion recombination events have efficiently taken place during the nanoplasma expansion, which require a copious number of quassifree electrons as well as ions in the system.

### 3.3. Ion TOF Spectrum of Xenon Clusters Irradiated with XFEL Pulses

In contrast to the electron spectrum, which mainly depicts the initial stage on the ionization dynamics of atomic cluster's ignition by the irradiation of hard-X-ray pulses, the ion TOF data is expected to depict the disintegration dynamics of nanoplasma in the latter stage. Figure 2a illustrates a contour map, where the horizontal and vertical axes represent the flight time and detection position of ions, respectively. In the present configuration of the X-ray beam and the detector, the ions with zero kinetic energy reached on the center of detector, $X \sim 0$ mm. The highly charged ions, $Xe^{q+}$ ($q = 4-15$), were detected around the center of detector, since these ions were produced only from the inner-shell photo-ionization of unbounded atoms contained in the cluster beam. The fact that the peaks of xenon isotopes are well separated (see an inset of Figure 2a) is further evidence that the kinetic energies of highly charged ions, $Xe^{q+}$ ($q = 4-15$), have been ~0 eV. At the present peak fluence (4.6 μJ/μm$^2$), the contribution of sequential multiple-photon absorption by an

atom [16] is negligibly small. In fact, the yield of $Xe^{15+}$ is almost invisible is Figure 2a. We note that the highest charge state is +14 due to the single photo-absorption of xenon atoms at 5.6 keV [53].

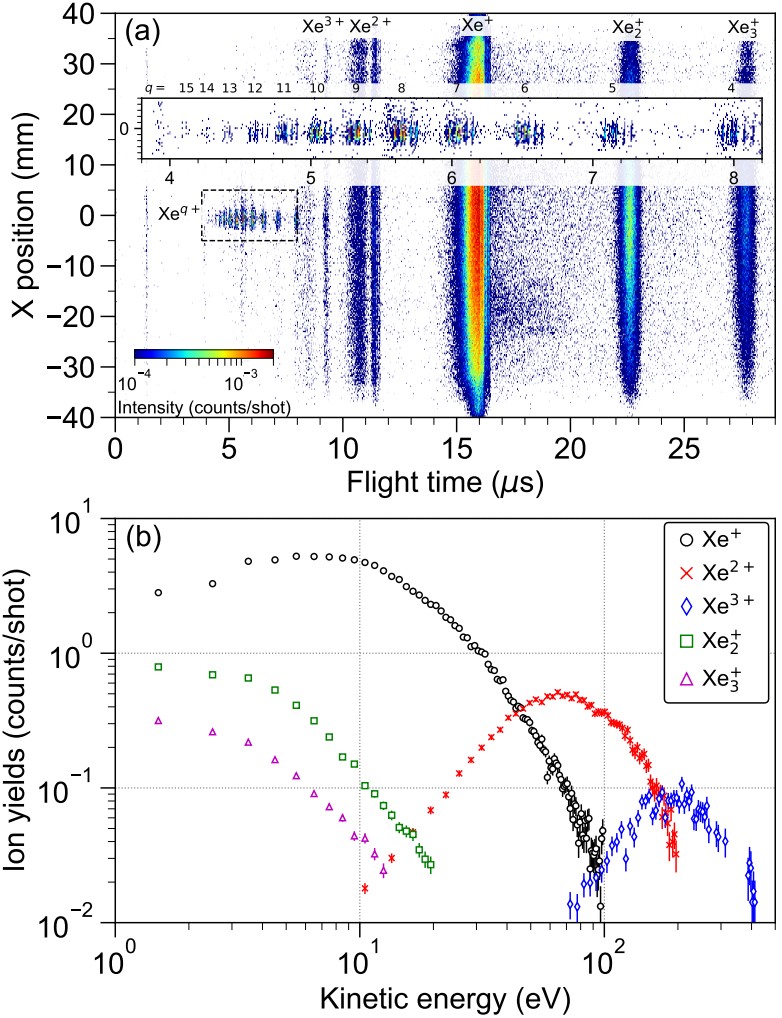

**Figure 2.** Ion TOF data obtained from $Xe_{5000}$ clusters at the XFEL peak fluence of 4.6 µJ/µm². (**a**) Flight time versus position $X$ maps. An inset depicts the magnified region of highly charged ions, $Xe^{q+}$ ($q = 4-15$). (**b**) Kinetic energy spectra of $Xe^+$, $Xe^{2+}$, $Xe^{3+}$, $Xe_2^+$, and $Xe_3^+$, respectively.

In Figure 2a, the images of $Xe^{q+}$ ($q = 1-3$) and $Xe_n^+$ ($n = 2, 3$) positions widely extend along the axis of the $X$ position. This result demonstrates that these ions have been produced via explosions of highly charged clusters, whereas the non-energetic ions, $Xe^{q+}$ ($q = 4-15$), have been produced from the unbounded atoms. The fact that the highly charged ions, $Xe^{q+}$ ($q \geq 4$), have not been produced from the ionized clusters means that the charges do not remain on the each core-ionized atoms, but promptly spread into the whole system before the ionized cluster expands.

The kinetic energy spectra of $Xe^+$, $Xe^{2+}$, $Xe^{3+}$, $Xe_2^+$, and $Xe_3^+$ are shown in Figure 2b. These kinetic energies are much smaller than those of hydrodynamic decay in NIR laser experiments [58–60]. In the previous report for argon clusters [30], the kinetic energy spectra of ionic oligomer fragments confirm that the disintegration dynamics of the XFEL-ionized clusters are strongly effected by the chemical processes, distributing the increasing net charge within the center part of the system and causing bond reorganization, which induces the formation of oligomer fragments. The kinetic energy spectra show that the singly charged ions, $Xe_n^+$ ($n = 1-3$), are mainly produced from the central part of the ionized

cluster. On the other hand, for the multiply charged ions, $Xe^{2+}$ and $Xe^{3+}$, the fraction of ions with less than 20 eV of kinetic energy account for several percent or less. This finding is presumed to result from most of the multiply charged ions being strongly accelerated during dissociation from the vicinity of the cluster surface, which was reported in the EUV experiments [61].

With the present peak fluence ($\sim 4.6$ μJ/μm$^2$), the ionization probability of individual xenon atoms is $\sim 8.7\%$; thus, $\sim 430$ atoms in the cluster of $\sim 5000$ atoms are core-ionized. Therefore, $\sim 400$ fast and $\sim 2900$ slow electrons are emitted from the core-ionized atoms in the cluster. Here, we simply assume that the slow electrons stay in the ionized cluster while the fast ones flay away from it. When a spherical object with the radius of 44 Å has the charge of +400, the electrostatic potential on its surface is 130 eV. As mentioned in Section 3.2, a number of the slow electrons escape from the charged clusters, and thus clusters' electrostatic potentials may become greater than 130 eV with the increasing charge. Again, the electrostatic potential on the surface of charged cluster provides the upper limit of the available electron temperature. Plasma electrons with the expected temperature cause the hydrodynamic expansion of nanoplasma that begins with surface ions. A fraction of outermost ions with high kinetic energy can expand fast enough to avoid electron capture.

Now we consider the ions with high kinetic energies that have been ejected from the vicinity of the charged cluster surface and assume that the XFEL-ignited nanoplasma has been constructed with a quasi-neutral core and a charged thin spherical shell [24,61–63]. In the charged thin spherical shell, the ions are expected to be accelerated by their own Coulomb repulsive forces. The potential energy of a singly charged ion at cluster's surface is $U_{\mathrm{Coul}} = \dfrac{e^2}{4\pi\epsilon_0}\dfrac{N_{\mathrm{eff}}}{R_0}$, taking into account the screening of ions by trapping potential [64]. Here, we consider that the charges have spread into the system already and define $e$ is the elementary charge, $\epsilon_0$ as the dielectric constant of the vacuum, $N_{\mathrm{eff}}$ as the effective number of unscreened charges on the cluster, and $R_0$ as the radius of the cluster. The $N_{\mathrm{eff}}$ contributing to the electrostatic potential is approximated by the difference between the total number of ion charges and the number of trapped electrons: $N_{\mathrm{eff}} = Q$. When the cluster size is 5000 atoms, the $R_0$ of the xenon cluster is 44 Å. When the ions are singly charged in the spherical shell, their mean kinetic energy corresponds to the energy stored per ion of the cluster's surface as $E_{\mathrm{kin}} = \left(\dfrac{1}{2}\dfrac{e^2}{4\pi\epsilon_0}\dfrac{N_{\mathrm{eff}}^2}{R_0}\right)/N_{\mathrm{eff}} = \dfrac{1}{2}U_{\mathrm{Coul}}$ [64]. We extended the use of this formula for the mean kinetic energy of the highly charged ions: $E_{\mathrm{kin,Xe}^{q+}} = \dfrac{1}{2}qU_{\mathrm{Coul}}$. When $Q$ is 400, the mean kinetic energies of $Xe^{2+}$ and $Xe^{3+}$ are $\sim 130$ and $\sim 200$ eV, respectively. The estimated mean kinetic energy of $Xe^{3+}$ agrees with the present observation (see Figure 2b); on the other hand, the estimated mean kinetic energy of $Xe^{2+}$ is slightly higher than the observed one. We note that the experimental results have been volume integrated [25,65], increasing the contribution from the lower fluence region of the beam focus. As will be discussed in the following Section 3.4, the low-fluence XFELs reduce $Q$ of the clusters and then provide ions with lower charges rather than higher ones. The low-fluence region of the XFEL beam's focus may be the main contributor to the creation of the $Xe^{2+}$ ions, and thus the observed kinetic energy becomes lower than the expected energy.

### 3.4. XFEL-Fluence Dependence

To discuss the fluence dependence of ionization dynamics for xenon clusters, we focus on the ion yields as a function of XFEL peak fluence. One can find two kinds of fluence dependence on singly ($Xe^+$, $Xe_2^+$, $Xe_3^+$) and multiply charged ($Xe^{2+}$, $Xe^{3+}$) ions, as shown in Figure 3a–e. In contrast, the yields of the multiply charged ions linearly increase with the XFEL fluence, while the yields of the singly charged ions become slightly saturated at a high peak fluence.

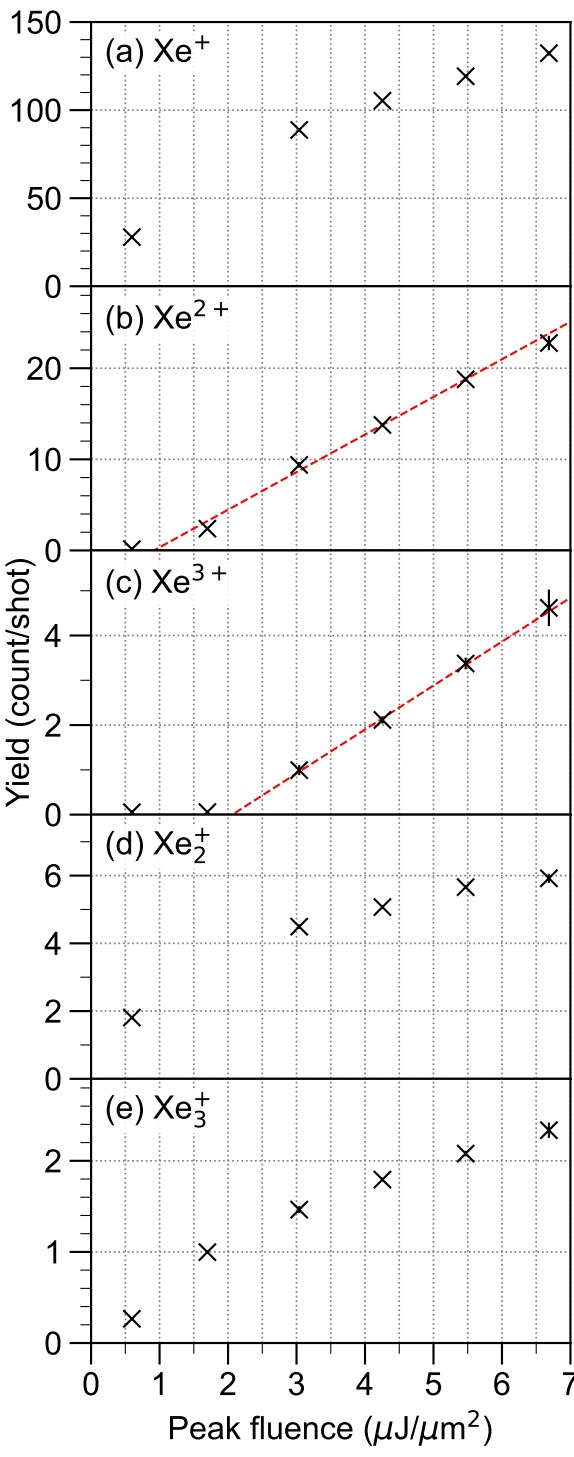

**Figure 3.** Ion yields as a function of XFEL peak fluence: (**a**) $Xe^+$, (**b**) $Xe^{2+}$, (**c**) $Xe^{3+}$, (**d**) $Xe_2^+$, and (**e**) $Xe_3^+$. Red broken lines are shown as guides for the eye in (**b**,**c**).

One can find that creating multiply charged ions calls for a minimum required fluence; for example, $Xe^{2+}$ and $Xe^{3+}$ ions have been observed above $\sim 0.9$ and $\sim 2.1$ $\mu J/\mu m^2$, respectively. Red dashed lines are shown as guides for the eye in Figure 3b,c. By considering that the charges in a cluster are redistributed to lower the stored Coulomb energy, the minimum required charge to produce one doubly charge ion on the thin spherical shell of xenon clusters has been estimated using the following formula: $Q_{\min,Xe2+} = \frac{5}{2}(I_{12} - I_{01})\frac{R_0}{e^2} - 4$ [66]. Here, $I_{01}$ and $I_{12}$ are the first and second ionization energies of xenon atoms, respectively.

When $R_0$ is 44 Å, $Q_{min,Xe^{2+}}$ is estimated to be ∼66. At the XFEL fluence of 0.9 μJ/μm$^2$, ∼80 fast and ∼580 slow electrons are emitted from the core-ionzied atoms in a xenon cluster with the size of 5000 atoms. With a simple assumption that the fast electrons of ∼80 can escape from the cluster system, the charges in the cluster at ∼0.9 μJ/μm$^2$ are greater than the minimum required charge to produce a doubly charged ion on the thin-spherical shell. In other words, when the XFEL fluence is less than the minimum required, the charges are not inhomogeneously distributed.

Let us now explore the fluence dependence of the yields of singly charged ions (Xe$^+$, Xe$_2^+$, Xe$_3^+$). The increase in $Q$ at a high XFEL fluence makes the charge distribution in a cluster even more inhomogeneous such that the thin-spherical shell has a larger number of charges. Therefore, even though $Q$ becomes high at high XFEL fluences, the proportion of the charge in the quasi-neutral core to the total charge decreases. Since the singly charged ions are mainly produced from the quasi-neutral core, their yields become saturated at the high peak fluence. Another reason is that the average charge of atoms in the cluster increases with the XFEL fluence. The reduction in the number of neutral atoms in the cluster inhibits the formation of chemical bonds between an atomic ion and the surrounding neutral atoms. Therefore, the saturation degree of Xe$_3^+$ is greater than that of Xe$^+$ and Xe$_2^+$.

*3.5. Cluster-Size Dependence*

Figure 4a shows the mean charge state of the detected ions (Xe$^+$, Xe$^{2+}$, Xe$^{3+}$, Xe$_2^+$, and Xe$_3^+$) as a function of cluster size at the XFEL peak fluence of ∼4.6 μJ/μm$^2$. Here, we have assumed that the charges of Xe$_2^+$ and Xe$_3^+$ are 1/2 and 1/3, respectively. The mean charge state of the detected ions for Xe$_{5000}$ is relatively higher than those for Xe$_{500}$ and Xe$_{1000}$. We note that the mean charge state has been increased by the formation of highly charge ions. Figure 4b depicts the percentages of Xe$^+$, Xe$^{2+}$, Xe$^{3+}$, Xe$_2^+$, and Xe$_3^+$ yields to total ion yield. A study using ion spectroscopy on xenon clusters with XUV pulses [66] reported that the doubly charged ions were only produced in the xenon clusters with average sizes of 10,000 and 50,000 but not 2000 atoms. This size dependence of the XUV laser can be explained by the minimum required charge to produce the doubly-charge ions on the thin spherical shell of xenon clusters [66]. Here, $Q_{min,Xe^{2+}}$, which is proportional to radii of clusters, is 28, 37, and 66 at Xe$_{500}$, Xe$_{1000}$ and Xe$_{5000}$ clusters, respectively. Since the number of fast electrons increasing the charge on a cluster are linearaly proportional to the number of core-ionized atoms, one can understand that in larger clusters, $Q$ is easily greater than the minimum required charge to produce doubly-charged ions. In other words, larger clusters easily form a quasineutral core and a charged thin spherical shell.

With the increase in the cluster size, the percentage of Xe$^+$ decreases, and the percentages of others increase, as shown in Figure 4b. The increase in the percentages of the multiply charged ions can be explained by the fact that, as mentioned above, with the increase in cluster size, the minimum required charge to produce multiply charged ions becomes smaller than the number of atoms in the cluster. The percentage of Xe$^{3+}$ at Xe$_{1000}$ is ∼1 %, which is close to at Xe$_{500}$; moreover, the percentage of Xe$^{2+}$ increases. On the other hand, the ratio of Xe$^{3+}$ become ∼2 % at Xe$_{5000}$. This fact can be understood by considering that $Q_{min,Xe^{3+}}$ is greater than $Q_{min,Xe^{2+}}$.

The percentages of the oligomer ions rapidly increase in comparison with the percentages of the multiply charged ions, e.g., the percentage of Xe$_3^+$ at Xe$_{5000}$ becomes about 10 times greater than at Xe$_{500}$. With the increase in cluster size, the charges are redistributed into the thin spherical shell, that is, the average charge of the atoms in the core is expected to be reduced. As discussed in Section 3.4, a sufficient number of neutral atoms may efficiently create chemical bonding with the atomic ions. We have experimentally demonstrated that the increasing cluster size effectively enhances the oligomer formation from the XFEL-ignited nanoplasma, and thus concluded that the chemical processes are essential for the disintegration dynamics of the larger-sized clusters.

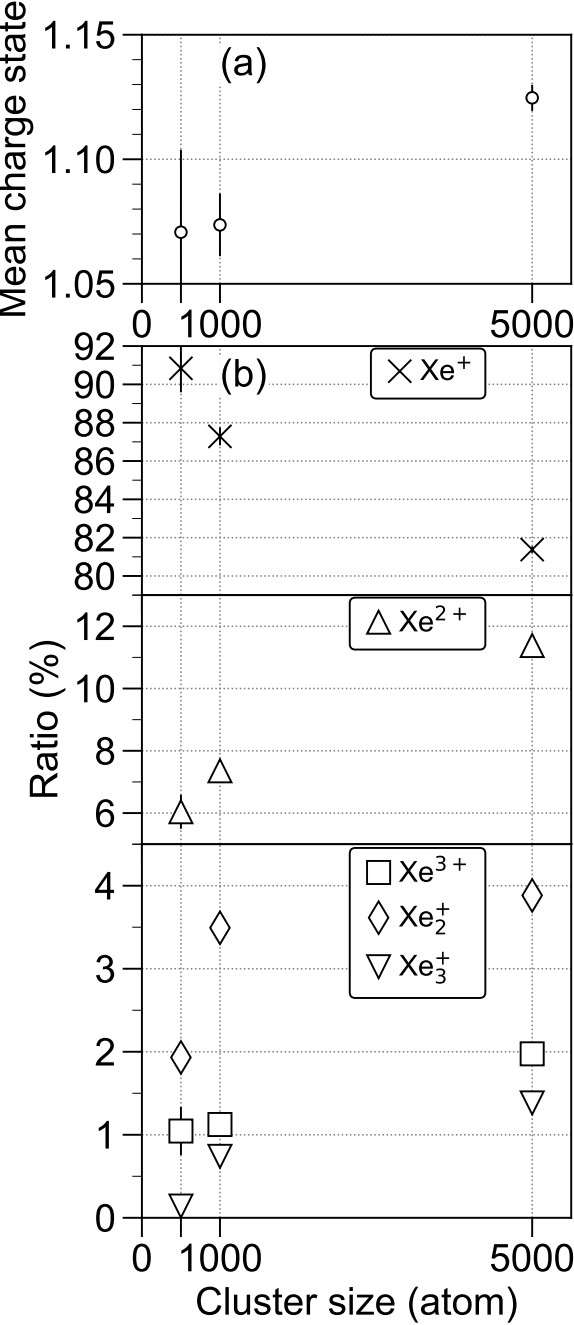

**Figure 4.** (**a**) Mean charge state of detected ions as a function of cluster size at the XFEL peak fluence of ~4.6 μJ/μm². (**b**) Percentages of Xe⁺, Xe²⁺, Xe³⁺, Xe₂⁺, and Xe₃⁺ to the total ion yield.

*3.6. XFEL Pump–NIR Probe Results*

Let us now focus on the results of XFEL pump–NIR probe experiments in order to investigate intermediate states of the expanding nanoplasma. Figure 5 shows the time-resolved electron energy spectra for xenon clusters with an average size of 5000 atoms, where the peak fluence of XFEL is ~1.8 μJ/μm² and the intensity of NIR laser is $5.0 \times 10^{12}$ W/cm². The red and black curves illustrate the electron energy spectra irradiated by an XFEL pump–pulse and a NIR probe–pulse with the time delays of +1.3 and −4.7 ps, respectively. The meaning of positive time delays is that the NIR probe–pulses arrive at the reaction point later than the XFEL pump pulses. At the present NIR laser intensity ($5.0 \times 10^{12}$ W/cm²), the Keldysh parameter is calculated to be 4.5, meaning that the multiphoton ionization is expected to

be dominant for individual xenon atoms. The present NIR laser intensity is too low to ionize xenon atoms via the multiphoton process; however, one can clearly observe that the additional irradiation with the weak probe laser at time delay of +1.3 ps causes a significant increase in the electron emission from the XFEL-ignited nanoplasma compared to the electron kinetic energy spectrum at −4.7 ps time delay. By subtracting the contribution of photo- and Auger electrons (depicted by the black curve in Figure 1b), the difference spectra have been obtained (see the blue and green curves in Figure 5). These difference spectra indicate that the NIR probe pulses with the positive time delays have mainly enhanced the unstructured contributions of electron spectra.

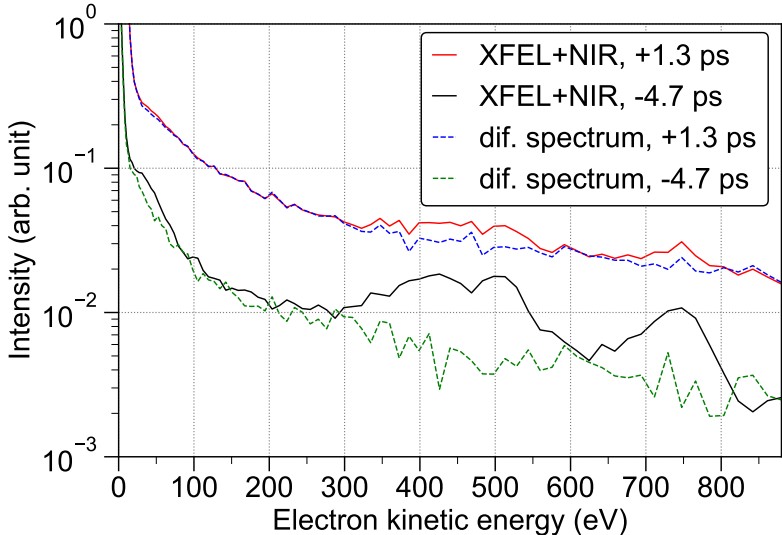

**Figure 5.** Electron energy spectra for xenon clusters with an average size of 5000 atoms measured at the XFEL peak fluence of $\sim$1.8 $\mu$J/$\mu$m$^2$ and the NIR intensity of $5.0 \times 10^{12}$ W/cm$^2$. Red and black curves depict the electron energy spectra at the time delays of +1.3 and −4.7 ps, respectively. Blue and green curves are the difference spectra obtained by subtracting the contributions of photo- and Auger electrons, shown as the black curve of Figure 1b, from the red and black ones, respectively.

The origin of the significant increase in the electron emission can be explained as follows. The quasi-free electrons within the nanoplasma can absorb energy from electric fields of the NIR laser [38,67]. The atoms and ions within the nanoplasma are further ionized by inelastic scattering of the heated-up quasifree electrons. As a result, not only the kinetic energy but also the yield of electrons emitted from the XFEL-ignited nanoplasma are significantly enhanced by being heated up with the NIR probe pulse. The decay constant of the exponential feature at the positive time delay has been increased to $\sim$60 eV (see the blue curve in Figure 5), which is an evidence that the NIR probe pulses have heated up the quasi-free electrons within the XFEL-ignited nanoplasma.

More evidence of the electrons heating-up with the NIR probe pulses is observed in the ion TOF spectra of xenon clusters with an average size of 5000 atoms, where the peak fluence of XFEL is $\sim$4.6 $\mu$J/$\mu$m$^2$ and the intensity of the NIR laser is $5.0 \times 10^{12}$ W/cm$^2$. The red and black curves in Figure 6a show the ion TOF spectra at the time delays of +1.5 and −1.5 ps, respectively. The enhancements in atomic ions, $Xe^+$, $Xe^{2+}$, $Xe^{3+}$, and $Xe^{4+}$, can also be clearly observed. The formation of $Xe^{4+}$, which was not observed at the negative time delay, suggests that the electrons heating-up with the NIR laser pulse has effectively increased the cluster charge $Q$. The increasing in $Q$ has increased the stored energy per ion of a cluster's surface, which results in a violent Coulomb explosion, producing ions with high kinetic energies (e.g., see the broaden peak of $Xe^{2+}$ on the red curve in Figure 6a). Moreover, one can find a diminished number of oligomer ions, $Xe_2^+$ and $Xe_3^+$, as an inverse phenomena for the increase in the yield of multiply charged ions at the positive time delay.

This finding can be ascribed to the binding being hindering by the reduction in the number of neutral atoms in the system with the increasing $Q$ due to the NIR heating up.

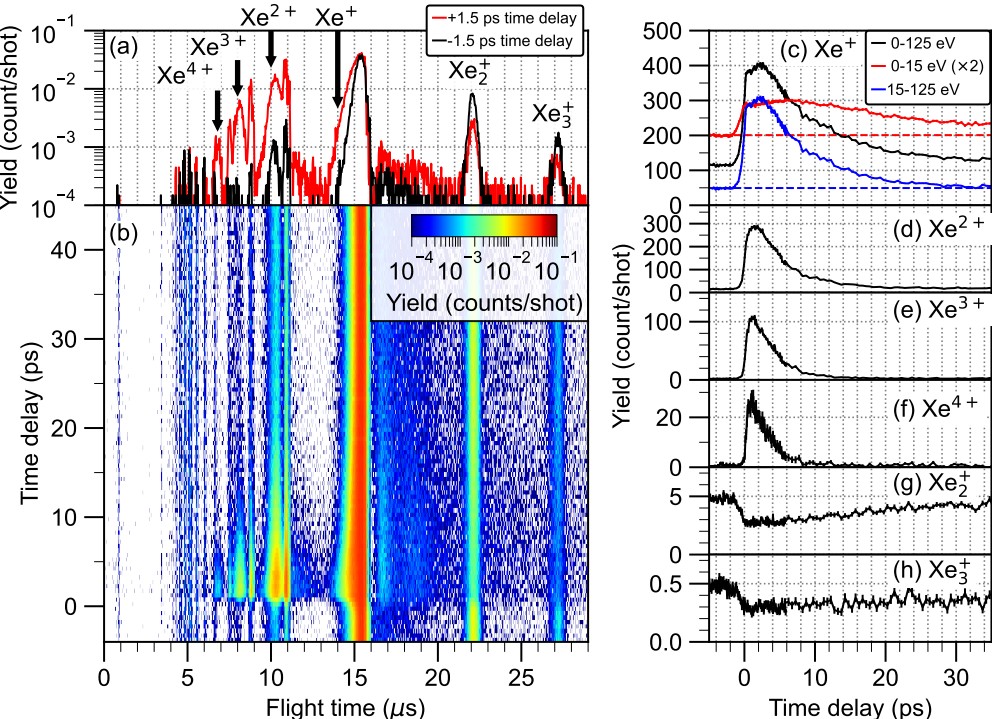

**Figure 6.** Ion TOF spectra for xenon clusters with an average size of 5000 atoms measured at the XFEL peak fluence of $\sim$4.6 $\mu$J/$\mu$m$^2$ and the NIR intensity of $5.0 \times 10^{12}$ W/cm$^2$. (**a**) Red and black curves depict ion TOF spectra at the time delays of +1.5 and $-1.5$ ps, respectively. The contributions of Xe$^+$, Xe$^{2+}$, Xe$^{3+}$, and Xe$^{4+}$ ions, which have been produced from the cluster ionization, are denoted with arrows. (**b**) Ion TOF spectrum as a function of time delay. Black curves of (**c**–**h**) depict ion yields of Xe$^+$, Xe$^{2+}$, Xe$^{3+}$, Xe$^{4+}$, Xe$_2^+$, and Xe$_3^+$, respectively, as a function of time delay. Red and blue curves in (**c**) show the yields of Xe$^+$ with low (0–15 eV) and high (15–125 eV) kinetic energies as a function of time delay, respectively. Red and blue dashed lines indicate the mean values of the red and blue solid curves at negative time delays, respectively.

Another effect of the NIR heating up can be observed on the kinetic energy of Xe$^+$. The mean kinetic energy of Xe$^+$ is $\sim$32 eV with the irradiation from NIR probe pulses at the time delay of +1.5 ps, which is about two times higher than the mean kinetic energy without the NIR-probe pulse. In contrast, the multiply charged ions are accelerated by the violent Coulomb explosion of the charged thin spherical shell, and the singly charged ions are mostly produced via a hydrodynamic expansion of a quasi-neutral core. As discussed with regard to the electron kinetic energy spectra measured by the pump–probe method, the additional NIR laser fields initially supply energy to the quasi-free electrons within the XFEL-ignited nanoplasma. Even if part of the absorbed energy of the quasi-free electrons must be expended by the further ionization of atoms and ions as well as the emission of heated-up electrons, the electron temperature becomes higher than the conditions without the NIR probe pulse. Since the plasma sound speed, which is the characteristic speed for hydrodynamic expansion, is proportional to the square root of the product of the electron temperature and the mean charge state of the nanoplasma [38], the formation of Xe$^+$ with the high kinetic energies has also evidenced that the XFEL-ignited nanoplasma has been heated up by the NIR probe pulses.

The ion TOF spectrum as a function of the time delay is illustrated in Figure 6b. One can find that the increase in the yield of multiply charged ions occurs up to $\sim$10 ps time

delay; on the other hand, the increase and decrease in the yield of singly charged ions persist until a time delay later than ~10 ps.

In order to discuss the time-delay dependence in more detail, we show the ion yields as a function of time delay in the black curves in Figure 6c–h. The yields of $Xe^{2+}$, $Xe^{3+}$, and $Xe^{4+}$ increased sharply from ~0 ps, reached peaks around 2 ps, and gradually decreased until ~15 ps. A similar time-delay dependence has been observed in the time-resolved ion spectroscopy by using the XUV pump–NIR probe method, which has been attributed to the surface plasma resonance effect [35].

Here, we consider the electron density, $n_e$, inside a spherical nanoplasma to satisfy the surface plasma resonance condition. Ditmire et al. described the surface plasma resonance effect for homogeneous nanoplasma [38], which can be mathematically tantamount to the Mie resonance. The heating efficiency of the surface plasma resonance depends on the surface plasma frequency, namely, the electron density inside a spherical nanoplasma. Surface plasma resonance is expected to occur when the surface plasma frequency, $\Omega_{pl} = \sqrt{\dfrac{e^2 n_e}{3\epsilon_0 m_e}}$, coincides with the probe-laser frequency, $\omega_{NIR}$. Here, $m_e$ is electron mass. The critical electron density, $n_{crit}$, has been defined to be $\dfrac{\epsilon_0 m_e \omega_{NIR}^2}{e^2} = 1.7 \times 10^{21}$ cm$^{-3}$. At the resonance condition $n_e = 3n_{crit} = 5.1 \times 10^{21}$ cm$^{-3}$, the heating efficiency in the cluster becomes enhanced.

Again, ~400 fast and ~2900 slow electrons are emitted from the individual core-ionized atoms in the xenon cluster with a size of 5000 atoms irradiated by the 5.5 keV photons with the peak fluence of ~4.6 μJ/μm$^2$. Assuming that ~400 fast electrons fly away from the ionized cluster system, the electron density inside the spherical nanoplasma is estimated to be ~$7.9 \times 10^{21}$ cm$^{-3}$ by the volume of pristine $Xe_{5000}$ ($3.7 \times 10^{-19}$ cm$^3$). We note that the electron density is expected to be greater than ~$7.9 \times 10^{21}$ cm$^{-3}$ as a results of the additional collisional ionization events with the trapped slow electrons, even though some of them escape from the nanoplasma.

When the electron density is much higher than $3n_{crit}$, the probe-laser field inside the cluster is screened by the dense quasi-free electrons of the nanoplasma. Thus, the laser field inside the cluster becomes weaker than the field on the outside. For instance, $n_e$ is much greater than ~$7.9 \times 10^{21}$ cm$^{-3}$ on the ionized clusters right after XFEL irradiation. Therefore, at ~0 ps time-delay the heating up of the nanoplasma is hindered by the screening effect. The electron density gradually decreases with time via thermal electron emission [26] from, and the electron–ion recombination within, the nanoplasma [24,37,57], as well as the expansion of nanoplasma [68–70]. When the resonance condition, $n_e = 3n_{crit} = 5.1 \times 10^{21}$ cm$^{-3}$, is fulfilled due to the evolution of the system on the picosecond timescale, the laser field can perforate into the XFEL-ignited nanoplasma and the heating efficiency becomes enhanced. Namely, the time-delay dependence features for the yields of the multiply charged ions must be attributed the time evolution of the expanding nanoplasma.

Let us now discuss the time evolution of electron density in the quasi-neutral core of XFEL-ignited nanoplasma that is hydrodynamically expanding. Here, we simply assume that the expansion speed of the quasi-neutral core is ~$3.5 \times 10^3$ m/s, which corresponds to the velocity of $Xe^+$ with the mean kinetic energy of ~17 eV. The quasi-neutral core expands to its radius of 52 Å at 200 fs time delay. Even if one excludes the reduction of quasi-free electrons due to the thermal electron emission and the electron–ion recombination, the electron density of a quasi-neutral core with a radius of 52 Å decreases to ~$5.1 \times 10^{21}$ cm$^{-3}$, reaching the critical electron density of $3n_{crit}$. The fact that the yields of multiply charged ions, $Xe^{2+}$, $Xe^{3+}$, and $Xe^{4+}$, culminate at the time delay later than 200 fs means that the quasi-free electrons have been effectively created via the collisional ionization events right after the XFEL irradiation, contributing to the formation of quasi-neutral core with a high electron density.

After the formation of peaks, the increments of $Xe^{2+}$, $Xe^{3+}$, and $Xe^{4+}$ gradually begin to decline with the decrease in the electron density of the expanding nanoplasma. On the other hand, the yields of oligomer ions, $Xe_2^+$ and $Xe_3^+$, are slowly restored at the time delays

later than ∼2 ps; in contrast, the yields of multiply-charged ions declined. We note that the surface plasma resonance effects cause sharp drops in the oligomer ion yields around 2 ps. However, the growth rate of the oligomer ions is much slower than the decline rate of the multiply charged ones. The difference of these rates has been ascribed to the fact that dispersed oligomer fragments are dissociated by the interaction with the weak NIR pulses [37,71]. The mean velocities of $Xe_2^+$ and $Xe_3^+$ have been estimated to be $\sim 1.8 \times 10^3$ and $\sim 1.2 \times 10^3$ m/s, respectively, considering their experimental mean kinetic energies of 4.6 and 3.1 eV. Even if we consider the occurrence of acceleration due to the electric fields supplied by the ion spectrometer, the ionic oligomer fragments remain in the focal spot of NIR probe pulses until ∼130 ns. Thus, the dissociation of dispersed oligomer fragments has persists until long time delays are reached.

The yield of $Xe^+$ is also effectively enhanced by the surface plasma resonance effects; however, its decline after ∼2 ps time delay is slower than that of the multiply charged ions. The slower decline might be caused by the formation of $Xe^+$ due to the oligomer dissociation caused by the NIR probe pulses. Now, we focus on the kinetic energy of $Xe^+$. The red and blue curves in Figure 6c depict the yields of $Xe^+$ with low (0–15 eV) and high (15–125 eV) kinetic energies, respectively. The shape of the blue curve in Figure 6c is similar to the curves of the multiply charged ions, indicating that the $Xe^+$ ions with kinetic energies of 15 to 125 eV have been produced due to the heating up of the XFEL-ignited nanoplasma by the NIR laser fields. On the other hand, oligomer dissociation caused by NIR laser fields was expected to provide slow $Xe^+$ ions, which is supported by the fact that the yield of $Xe^+$ with kinetic energies of up to 15 eV was still enhanced at the long time delay of ∼30 ps. Moreover, another effect to increase slow $Xe^+$ ions may be the reionization of highly excited atoms produced by electron–ion recombination events during the nanoplasma expansion [72].

Let us now focus on the gradual declines in the yield of the multiply charged ions after their peaks in order to discuss the electron distribution in the XFEL-ignited nanoplasma. With the assumption, as mentioned above, that the expansion speed of nanoplasma is $\sim 3.5 \times 10^3$ m/s, at ∼5 ps time-delay the electron density decreases to ∼1 % of the initial one, even if the thermal electron emission and the electron–ion recombination have been excluded. When the quasi-free electrons are homogeneously distributed in the nanoplasma, the heating efficiency of the nanoplasma with a low electron density is equal to that of IBS heating. However, the IBS heating is not expected to supply sufficient energy in the XFEL-ignited nanoplasma for creating the highly charged ions (e.g., $Xe^{4+}$).

Recent molecular dynamics simulations reported the collective excitation modes of a nanoplasma with an inhomogeneous electron density [73–75]. The resonance frequency of an inhomogeneous electron system becomes lower than the Mie resonance frequency, where the electron density of the nanoplasma is assumed to be constant [73]. The resonance frequency tends to be a Mie frequency with the increase in the cluster size [73,74]. In other words, the surface plasma resonance for the nanoplasma with an inhomogeneous electron system occurs at a lower electron density than a homogeneous one.

The Mie-like resonance becomes much broader due to the motion in the fluctuation field of point-like ions [75]. This broadening must expand the range of electron density for causing the effective heating of nanoplasma. Therefore, if the electron density is inhomogeneous, effective heating occurs on the XFEL-ignited nanoplasma with an electron density lower than the critical electron density. Thus, we conclude that the gradual declines in the yield of multiply charged ions are attributed to the XFEL-ignited nanoplasma with a finite size having inhomogeneous electron density. Moreover, as mentioned above, the inhomogeneous electron density makes the resonance electron density lower than Mie resonance, which shifts the peaks of the multiply charged ions to a time delay later than what was expected. This finding is consistent with the experimental observations. A more detailed investigation using molecular dynamics simulations is necessary to quantitatively analyze the ion yield curves for the XFEL pump–NIR probe experiments because the molecular dynamics simulations address the real space dynamics of atoms, atomic ions,

and electrons, including electron impact ionization, inelastic scattering, as well as electron-ion recombination processes. Moreover, the two-step molecular dynamics–hydrodynamic approach is expected to track the expansion of the quasi-neutral nanoplasma core in the longer picosecond timescale [76,77].

The total energy absorption of the cluster irradiated by the XFEL and NIR pulses can be determined as the sum of the kinetic energies of the emitted electrons and ions, of the ionization potentials required to produce all ions, and of the energies of the photons emitted from electronically excited atoms. In the present study, the sum of kinetic energies of emitted ions and the ionization potentials required to produce all ions read as $S_{\text{EI}} = \sum_{\text{ion}} \int E_{\text{kin,ion}} Y_{\text{ion}}(E_{\text{kin,ion}}) dE_{\text{kin,ion}}$ and $S_{\text{IP}} = \sum_{\text{ion}} IP_{\text{ion}} \int Y_{\text{ion}}(E_{\text{kin,ion}}) dE_{\text{kin,ion}}$, respectively, where $E_{\text{kin,ion}}$ is the kinetic energy of the ion, $Y_{\text{ion}}(E_{\text{kin,ion}})$ is the ion yield with the kinetic energy of $E_{\text{kin,ion}}$, and $IP_{\text{ion}}$ is the ionization potential of the ion. $IP_{\text{ion}}$ can be accurately evaluated from the atomic ionization potentials available in the literature [78]. Assuming that the ion system is in good thermal equilibrium with the electrons system, the total energy absorption is propotional to the sum of $S_{\text{EI}}$ and $S_{\text{IP}}$.

Figure 7 shows the relative total energy absorption ($S_{\text{EI}} + S_{\text{IP}}$) of xenon clusters with an average size of 5000 atoms irradiated by the XFEL and NIR pulses as a function of the strength of the NIR-laser field, $|E_0|$, where the peak fluence of XFEL pulse is $\sim$4.6 μJ/μm$^2$ and the time delay is 2 ps. The heating efficiency under this resonance condition is proportional to $\nu^{-1}|E_0|^2$, where $\nu$ is the electron–ion collision frequency [38]. The electron–ion collision frequency is proportional to $|E_0|^{-3}$ according to the standard Coulomb formulas [79]. One can clearly see that the relative total energy absorption is proportional to $|E_0|^5$ in Figure 7, which has also provided evidence that heating up the XFEL-ignited nanoplasma using an NIR-laser pulse can be ascribed to the the surface plasma resonance.

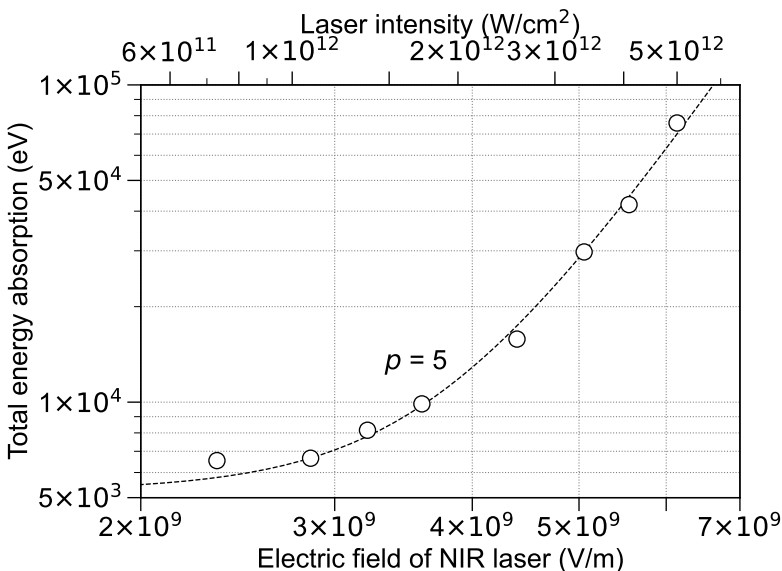

**Figure 7.** Total energy absorption of xenon clusters with an average size of 5000 atoms irradiated by the XFEL and NIR pulses as a function of the strength of the NIR-laser field. The peak fluence of XFEL pulse is $\sim$4.6 μJ/μm$^2$ and the time delay is 2 ps. The electric field to the fifth power is shown as a dashed curve to guide the eye.

## 4. Conclusions

We investigated the ionization dynamics of xenon clusters irradiated with a hard X-ray laser pulse with the photon energy of 5.5 keV by using electron and ion spectroscopies. The kinetic energy spectra of the ion fragments ejected by xenon clusters with an average size of 5000 atoms have shown that multiply charged ions with high kinetic energies are ejected from the vicinity of the cluster's surface. Since the charges in the cluster are redistributed

to reduce the stored Coulomb energy, the XFEL-ignited nanoplasma is formed with a quasi-neutral core and a charged thin spherical shell, a finding which has been reported by both XUV [61] and soft X-ray experiments [62].

The dependence of ion yields on the XFEL fluence has evidenced the minimum charge required on the nanoplasma to produce multiply charged ions from the thin spherical shell of the clusters. The dependence of the yields of singly charged ions on the XFEL fluence indicates that the oligomer formation due to the bonding reorganization is disturbed by increasing the mean charge state of the ions within the XFEL-ignited nanoplasma. The percentages of oligomer ions to the total ion yields being a function of the cluster size suggests that increasing the cluster size enhances the bonding reorganization. Consequently, we have experimentally demonstrated the importance of XFEL-fluence and the object's size in achieving control over the bonding reorganization to create oligomer fragments.

Using xenon clusters with an average size of 5000 atoms, we have conducted time-resolved spectroscopy experiments using delayed NIR laser pulses using a wavelength of 800 nm as a probe. The XFEL-ignited nanoplasma was heated up by the surface plasma resonance when the electron density of expanding nanoplasma met the resonance condition. The gradual decline in the yield of the multiply-charged ions after the peaks were reached might suggest that the quasi-free electrons are inhomogeneously distributed within the XFEL-ignited nanoplasma.

**Author Contributions:** H.F., K.N., and K.U. conceived the experiment; K.U. coordinated the experimental team; W.X., T.H.N., K.M. (Koji Motomura), S.-i.W., S.M., T.T. (Tetsuya Tachibana), Y.I., T.S., K.M. (Kneji Matsunami), T.U., C.N., H.F., K.N., and K.U. set up the experiment; T.T. (Tadashi Togashi), K.O., and S.O. prepared the NIR laser; W.X., T.H.N., K.M. (Koji Motomura), S.-i.W., S.M., T.T. (Tetsuya Tachibana), Y.I., T.S., K.M. (Kneji Matsunami), T.U., C.N., C.M., T.T. (Tadashi Togashi), K.O., S.O., K.T., M.Y., H.F., K.N., and K.U. conducted the experiment; Y.K., W.X., K.A., T.H.N., K.M. (Koji Motomura), D.I. analyzed the data; Y.K., W.X., K.A., C.M., H.F., K.N., and K.U. interpreted the data and wrote the paper. All authors have read and agreed to the published version of the manuscript.

**Funding:** This study was supported by the X-ray Free Electron Laser Utilization Research Project and the X-ray Free Electron Laser Priority Strategy Program of the Ministry of Education, Culture, Sports, Science, and Technology of Japan (MEXT); by the Proposal Program of SACLA Experimental Instruments of RIKEN; by the Japan Society for the Promotion of Science (JSPS) KAKENHI Grant No. JP21244042, No. JP23241033, No. JP15K17487, No. JP16K05016; by MEXT KAKENHI Grant No. JP22740264; by the IMRAM project; and by the National Nature Science Foundation of China (Grant No. 12174259). H.F. and K.U. acknowledge "Dynamic Alliance for Open Innovation Bridging Human, Environment, and Materials". S.W. and K.N. acknowledge the Research Program "Dynamic Alliance for Open Innovation Bridging Human, Environment and Materials" in "Network Joint Research Center for Materials and Devices". S.M. acknowledges JSPS KAKENHI Grant No. JP11F01028. T.H.N. acknowledges the Research Program for Next Generation Young Scientists "Dynamic Alliance for Open Innovation Bridging Human, Environment and Materials" in "Network Joint Research Center for Materials and Devices".

**Institutional Review Board Statement:** Not applicable.

**Informed Consent Statement:** Not applicable.

**Data Availability Statement:** The data that support the findings of this study are available upon reasonable request from the authors.

**Acknowledgments:** We are grateful to the late Makoto Yao for his invaluable contributions to the present work. The experiments were performed at SACLA with the approval of JASRI and the program review committee (No. 2014A8040). D.I. and Y.I. acknowledge IMRAM, Tohoku University.

**Conflicts of Interest:** The authors declare no conflict of interest.



## Abbreviations

The following abbreviations are used in this manuscript:

| | |
|---|---|
| XFEL | X-ray free electron laser |
| NIR | Near-infrared |
| IBS | Inverse Bremsstrahlung |
| XUV | Extreme ultraviolet |
| FWHM | Full width at half maximum |
| KB | Kirkpatrick–Baez |
| TOF | Time-of-flight |
| VMI | Velocity map imaging |

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
