# Peer review of "Ionization of Xenon Clusters by a Hard X-ray Laser Pulse"

_applsci, doi:10.3390/app13042176_

Round 1

Reviewer 1 Report

The manuscript "Ionization of xenon clusters by a hard x-ray laser pulse" by Y. Kumagai et al. reports on pump-probe experiments of Xe clusters. The pump pulse consisted of a x-ray pulse of 5.5keV photon energy, the probe pulse of a weak NIR pulse of 5´10‑12 Wcm2 with a temporal length of 82fs, not being able to ionize Xe by multiphoton or tunnel ionization without prior to the x-ray pulse. The average cluster size was 5000 atoms. The authors present also pump-only results.

All in all, the manuscript contains very nice new experimental results. I therefore recommend to publish the manuscript after minor improvements/clarifications. I have the following comments:

(1) In lines 297-298 hydrodynamic expansion is assumed as the mechanism for the ion acceleration, but in lines 304 and 312 Coulomb repulsion is used to calculate ion kinetic energies. This is inconsistent in my opinion.

(2) In line 312 I do not understand the factor 1/2 in the calculation of the surface ion kinetic energy. The factor 1/2 means that the kinetic energy available from the Coulomb repulsion of the surface ion of charge q and the rest of the cluster of net charge Q is equally partitioned among the ion and the remaining cluster body. However, because of the very unequal masses the surface ion should attain nearly all the available kinetic energy.

(3) Since the authors use the hydrodynamic expansion model a lot for qualitative interpretations, I would like to bring two papers of Last et al. to their attention: Adv. Quant. Chem. 75, 27 (2017), and Mol. Phys. 116, 2461 (2018). In these two papers, the validity of the hydrodynamic expansion model is questioned. The authors might put their qualitative interpretations in perspective accordingly, if they like.

(4) The manuscript contains a considerable number of misprints and some mishaps, for example

Line 434: "at the pump-probe method"  Replace "at" by "by".

Line 454: "we consider about the electron density"  Take out "about".

Reviewer 2 Report

The authors investigated the behaviour of the nanoplasma created by a hard x-ray pulse interacting with xenon clusters by using electron and ion spectroscopy and they present the time-resolved data revealed by a near-infrared (NIR) probe pulse. The manuscript is well prepared, the results and the experiment, and all figures are properly addressed.

Reviewer 3 Report

This work is devoted to studying of Ionization of xenon clusters by a hard x-ray laser pulse. Authors confirmed that The gradual decline of the multiply-charged ions yields after the peaks might suggest that the quasifree electrons are inhomogeneously distributed within the XFEL-ignited nanoplasma.

I advise the authors to take the following points into account while revising their manuscript.

1-Check the manuscript for grammatical errors

2- Abstract needs to be improved, highlighting the problem statement and the current study approach.

3-Paper organization: generally good, but the paper would benefit from another thorough proofreading. I was occasionally distracted by incorrect or missing abbreviations.

4- References are written differently in work (try to add DOI for all ref such as ref.52 DOI 10.1088/0953-4075/29/16/008)

Reviewer 4 Report

In this manuscript, an experimental approach is followed to ionize xenon clusters. In general, the paper seems to be scientifically accurate and novel. Additionally, the topic is a very hot one in the atmosphere of optics. As the results are well interpreted and expanded to different situations, the reviewer does not see any major problem with it. However, some minor issues must be addressed by the authors in detail. First off, more explanation about modifications made possible with the aid of nanotechnology in this area is demanded. Secondly, in spite of the fact that current work is novel, no good bridge was made by the authors between previous works and theirs. It is necessary to prove the novelty of what you have reported in your article. In the third place, it would be very great if the authors provide some practical insight about potential application of their study in the battery industry.
